# A Simple Vortex-Based Method for the Generation of High-Throughput Spherical Micro- and Nanohydrogels

**DOI:** 10.3390/ijms26136300

**Published:** 2025-06-30

**Authors:** Moussa Boujemaa, Remi Peters, Jiabin Luan, Yieuw Hin Mok, Shauni Keller, Daniela A. Wilson

**Affiliations:** Institute for Molecules and Materials, Radboud University, Heyendaalseweg 135, 6525 AJ Nijmegen, The Netherlands; moussa.boujemaa@ru.nl (M.B.); remi.peters@ru.nl (R.P.); j.luan@science.ru.nl (J.L.); shauni.keller@ru.nl (S.K.)

**Keywords:** vortex emulsification, inverse-emulsion polymerization, PEGDA, water-in-oil, microgels, nanogels

## Abstract

Hydrogel particles, renowned for their high water content and biocompatibility in drug delivery and tissue engineering, typically rely on complex, costly microfluidic systems to reach sub 5 µm dimensions. We present a vortex-based inverse-emulsion polymerization strategy in which UV crosslinking of polyethylene glycol diacrylate (PEGDA) dispersed in n-hexadecane and squalene yields tunable micro- and nanogels while delineating the parameters that govern particle size and uniformity. Systematic variation in surfactant concentration, vessel volume, continuous phase viscosity, vortex speed and duration, oil-to-polymer ratio, polymer molecular weight, and pulsed vortexing revealed that increases in surfactant level, vortex intensity/duration, vessel volume, and oil-to-polymer ratio each reduced mean diameter and PDI, whereas higher polymer molecular weight and continuous phase viscosity broadened the size distribution. We further investigated how these same parameters can be tuned to shift particle populations between nano- and microscale regimes. Under optimized conditions, microhydrogels achieved a coefficient of variation of 0.26 and a PDI of 0.07, with excellent reproducibility, and nanogels measured 161 nm (PDI = 0.05). This rapid, cost-effective method enables precise and scalable control over hydrogel dimensions using only standard laboratory equipment, without specialized training.

## 1. Introduction

Hydrogels are typically defined as cross-linked three-dimensional polymer networks in which water serves as a swelling agent, conferring versatile properties such as high water content and biocompatibility that render them ideal for numerous biomedical applications [1]. These applications range from controlled drug delivery [2,3] and tissue engineering [4,5,6] to gene therapies [7] and tumor immunotherapies [8,9]. Micro- and nanogels, in particular, are well-suited for drug delivery due to their enhanced surface-to-volume ratio, which increases their efficacy in targeted therapeutic applications [10]. Notably, nanogels exhibit improved cellular uptake [11], enhanced and non-invasive drug delivery [12], and improved tissue penetration [13], thereby further optimizing their performance as carriers for therapeutic agents.

Various hydrogel preparation techniques are well established in the literature. These methods include, but are not limited to, electrospinning [5], spray drying [14], sol–gel transition [15], mold casting [16], and cryo-gelation [17]. Among these, microfluidics [18,19,20] stands out as the most commonly employed technique for generating hydrogel particles.

Microfluidics allows for a precise manipulation of fluids at the microscale, enabling the high-throughput production of monodisperse droplets with diverse compositions and morphologies [21]. The generation of hydrogels via microfluidics often utilizes either glass capillary devices [22] or polydimethylsiloxane (PDMS) chips [23,24], the latter being fabricated via photolithography to create intricate channel patterns, which are then bonded to a substrate, usually glass, to form enclosed micro-channels [25]. Glass capillary devices, however, are typically larger in scale, meaning they can only produce relatively large hydrogel droplets; for the generation of smaller hydrogel particles, PDMS-based microfluidic chips are generally required. While microfluidics excels at generating uniform droplets, it can be time-consuming and involves multiple setup steps, from mold design and layer alignment to chip fabrication, all of which demand specialized expertise and dedicated laboratory space [26]. Furthermore, the equipment and materials, such as micropumps, microscopes and PDMS, significantly increase costs, underscoring the need for simpler and more cost-effective droplet generation technologies.

Various techniques have been reported for the synthesis of micro- and nanogels without relying on microfluidics. These include spray drying [27], electrohydrodynamic spraying [28,29], and particle-templated emulsification [30,31] for microgels, as well as radical polymerization [32,33], micro-emulsification [34,35,36], and inverse mini-emulsion polymerization [37] for nanogels. While effective, many of these approaches require elevated temperatures or prolonged synthesis times, which limits their scalability and practicality. Emulsification using a simple vortex mixer has emerged as a more accessible alternative, enabling rapid droplet generation within minutes. This method can also produce complex structures such as core–shell [38] and Janus structures [39], previously achievable only with complex microfluidic chip designs [40]. However, while vortex-based emulsification simplifies production, it remains limited in achieving small droplet sizes and precise polydispersity control, falling short of microfluidic droplet uniformity.

In this study, we introduce a novel and straightforward method for generating small micro- and nanogels, using polyethylene glycol diacrylate (PEGDA) as a model polymer, employing a simple and cost-effective vortex system. By exploring the effects of different parameters on droplet size and polydispersity, we demonstrate how this approach can consistently produce uniform sub-5-micron hydrogels, offering a faster alternative to traditional microfluidic methods.

## 2. Results and Discussion

Our technique employs a single-step, microfluidic-free emulsification platform to produce small, uniform hydrogel particles by simple pipetting and controlled vortexing. As illustrated in the schematic in Figure 1, PEGDA in ultrapure water was used as a model dispersed phase, while the continuous phase consisted of n-hexadecane oil with Span^®^80 as the surfactant. To prevent premature emulsification, both solutions were carefully introduced into the Eppendorf^®^ tube before vortexing. Upon vortexing, the shear forces at the tube wall dispersed the aqueous phase into high-throughput water-in-oil (w/o) droplets [41], which were then polymerized under a focused UV beam and washed to yield spherical hydrogel microparticles (Appendix A).

Because vortexing inherently produces a broad droplet size distribution spanning the nano- to microscale, we systematically investigated key variables that affected the mean size and polydispersity index (PDI) of the formed hydrogel distribution. By tuning these parameters and applying simple post-processing (e.g., centrifugation), we further demonstrate the ability to isolate and collect exclusively micro- or nanoscale hydrogel populations, thus offering a versatile, high-throughput route to tailored particle synthesis with only standard laboratory equipment and no specialized training.

Prior to characterization, the hydrogel suspension was fractionated by centrifugation: an initial spin at 2000 rpm for 5 min sedimented the microgel fraction, leaving the nanogels in the supernatant for subsequent analysis. The resulting pellets containing the microgels were further separated into two different populations (Appendix A).

### 2.1. Variable Effects on Microgels

The key variables initially anticipated to significantly influence the formed emulsion droplets included surfactant concentration, vortex duration, vortex speed, polymer molecular weight, oil-to-polymer ratio, and Eppendorf^®^ size. Each factor was systematically varied to assess its impact on particle size and PDI, enabling us to determine the optimal conditions for producing small, uniform hydrogel particles. Each variable was first evaluated at two contrasting settings (e.g., high vs. low concentration or short vs. extended duration) to initially assess its principal effect. Once the effect was validated, additional levels were introduced to capture trends and refined analysis.

Several variables were found to considerably influence the average size and PDI of the formed microgels. The reduction in emulsion droplet size from 4.89 µm to 3.32 µm, a 32% decrease with increasing surfactant concentration, can be attributed to an improved interfacial coverage by surfactant molecules [42], resulting in a further reduction in interfacial tension [43]. This in turn facilitated the breakup of larger droplets into smaller ones during the emulsification process (Figure 2A). The increased availability of surfactant molecules also stabilized smaller droplets by creating a strong barrier against coalescence, which improved emulsification efficiency and ensured droplet stability [44].

The effect of Eppendorf^®^ size can be attributed to the enhanced uniformity of the shear gradient. Larger Eppendorf^®^ sizes provided more space for fluid flow, enabling a more even distribution of shear forces compared to smaller sizes. Furthermore, enlarging the Eppendorf tube increased the vessel’s radius (r). In the steady state of rigid-body rotation—where fluid particles move together without slip—the tangential speed isu*_θ_*(r) = Ω r(1)
where Ω is the constant angular velocity and u*_θ_* is the tangential velocity at radius r [45,46]. Higher tangential velocities generate stronger shear forces on suspended particles, causing them to break apart or mix more thoroughly and thus reducing their effective size. Because the angular velocity, i.e., the vortex speed, is fixed, increasing the vessel’s radius directly raises the tangential velocity. In turn, a larger r yields a higher u*_θ_* at the wall, intensifying shear and resulting in smaller particles under otherwise identical conditions. Notably, a 29% decrease in mean particle diameter from 4.58 µm to 3.25 µm was observed when increasing the Eppendorf^®^ size from 0.5 mL to 5 mL. This reduction in droplet size was also accompanied by a significant narrowing of the size distribution, indicating improved uniformity within the emulsion (Figure 2B).

Similarly, increasing the vortex speed and duration generated stronger and more sustained shear forces within the solution, offering additional time for droplet breakup and leading to improved droplet size reduction and uniformity, i.e., 8.19 µm to 4.25 µm (−48%) and 4.46 µm to 3.32 µm (−26%), respectively (Figure 2C,D) [38,44,47]. However, at lower vortexing speeds, an incomplete emulsification of the dispersed phase was observed, resulting in a lower yield of hydrogel particles (Appendix A) [38].

Finally, increasing the oil-to-polymer ratio led to smaller droplets while also reducing the dispersity index, resulting in a more uniform size distribution. Increasing the volume of the continuous phase added more surfactant molecules to the system, which enhanced the stabilization of formed emulsion droplets (Figure 2E). Concurrently, a larger continuous phase volume increased the distance between dispersed droplets, reducing the likelihood of coalescence. Conversely, a reduction in the oil-to-polymer ratio led to an increase in mean droplet size, i.e., 3.41 µm to 3.67 µm (8% increase). This effect arose from a decrease in surfactant concentration and a reduced spacing between emulsion droplets, which undermined droplet stabilization and promoted coalescence (Figure 2F).

We also identified a critical limitation related to the size of the container used for emulsion generation. Specifically, when the emulsion reached the cap of the vial during vortexing, the emulsification process became inefficient. This inefficiency occurred because, when the vial was overfilled, the emulsion solution could not be pressed thinly against the vial walls. Instead, the solution remained thicker along the sides of the vial, reducing the effectiveness of the shear forces required to break up larger droplets into smaller ones (Appendix A). This observation underscores the importance of selecting an appropriately sized container to ensure optimal emulsification efficiency. The observed increase in droplet size and dispersity at the highest oil-to-polymer ratios (50:600 and 50:800) in Figure 2E could be attributed to this effect. Specifically, increasing the oil volume from 50 µL to 400 µL resulted in a 37% reduction in mean particle diameter, from 5.9 µm to 3.72 µm. However, further increasing the oil volume to 800 µL led to a 22% increase in mean particle diameter, from 3.72 µm to 4.52 µm. These findings highlighted the critical interplay between vessel size, shear forces, and emulsion quality.

Similarly, a change in polymer molecular weight (MW) also affected particle uniformity (Figure 3A). In microfluidic systems, increasing polymer MW resulted in an increased dispersed phase viscosity, which inhibited the breakup process and promoted the formation of larger droplets [48,49,50]. Comparably, as shown in Figure 3A and Appendix A, increasing PEGDA MW raised the dispersed phase viscosity, amplifying resistance to droplet deformation and resulted in a broader size distribution.

Finally, a novel approach was identified to further reduce the size of the emulsion droplets, namely vortexing in pulses. In this protocol, an Eppendorf^®^ tube is vortexed for a preset interval (e.g., 5, 10, or 30 s) and then removed and allowed to rest for 2 s before being vortexed again for the same duration. This cycle of mixing and brief pauses is repeated for the whole vortexing duration. Notably, a steady decrease in mean droplet diameter was observed as pulse interval duration increased, with a significant size reduction achieved using 30 s pulses, i.e., 3.82 µm to 3.23 µm (15% decrease) (Figure 3B). Furthermore, modest improvements in emulsion dispersity were observed. The emulsification process during vortexing is governed by two competing phenomena: droplet breakup and droplet coalescence. Breakup occurs when vortex-induced turbulence and shear forces fragment larger droplets into smaller ones, while coalescence results from increased droplet collisions that promote the merging of smaller droplets into larger ones. The balance between these opposing processes ultimately determines the final droplet size distribution. As vortexing continues, a dynamic equilibrium is typically established, stabilizing the emulsion structure over time [51]. We speculate that pulsatile vortexing disrupts this equilibrium by introducing brief relaxation periods between pulses. At the onset of each pulse, the emulsion transitions rapidly from rest to motion, generating steep velocity gradients and thereby inducing high shear forces. Repeating this transition multiple times, as opposed to once, may enhance droplet breakup beyond what is achieved with continuous agitation. Simultaneously, the intermittent pauses may reduce droplet collision frequency, limiting coalescence.

Moreover, each vortexing pulse likely disturbs the evolving flow field within the emulsion, preventing the formation of stabilized flow patterns that typically develop under continuous vortexing. These repeated flow disruptions, combined with alternating shear and relaxation phases, are likely key contributors to the observed reduction in mean droplet diameter (Figure 3B). While direct measurements of turbulence characteristics or droplet count kinetics were not performed, these mechanisms offer a plausible explanation for the observed trends. Additionally, pulse vortexing could amplify cavitation within the emulsion, further contributing to the observed reduction in mean hydrogel size [52].

As illustrated in Figure 3C, the proposed vortex-based emulsification technique demonstrates excellent reproducibility, showing similar size distributions and no significant difference in mean diameter when performed in triplicate. This establishes the proposed method as a reliable and consistent alternative to microfluidics for generating small and uniform hydrogel microparticles (Figure 3D) while also underscoring the efficiency of our protocol as a faster and viable approach for hydrogel generation. Although the vortex-based method produced a coefficient of variation (C.V.) less optimal than microfluidics, our work demonstrated that it can be refined to achieve a C.V. of 0.26 with a corresponding PDI of 0.07 (Appendix A).

### 2.2. Variable Effect on Nanogels

To characterize trends within the nanogel fraction, the supernatant—previously separated from the microgel fraction by centrifugation—was further purified by spinning at 13,300 rpm for 5 min. We then assessed the same key variables, with the addition of crosslinking density, to determine whether nanogels exhibited analogous behaviors. Since pulsatile vortexing and larger Eppendorf^®^ tubes outperformed their counterparts, all samples were prepared in 5 mL tubes using 30 s vortex pulses.

Increasing surfactant concentration from 1% to 10% produced a 40% reduction in PDI, comparable to that previously observed for microgels, and enhanced the overall nanogel yield (Figure 4A, Appendix A). This improvement reflects the greater abundance of surfactant molecules lowering the interfacial tension of the water-in-oil emulsion. At 10% surfactant, simple centrifugation produced a PDI as low as 0.05, clearly demonstrating its effectiveness in purifying a very narrow size distribution (Appendix A). Notably, the mean nanogel diameter increased by 8% between 5 and 10% of Span^®^ 80, increasing from 157 nm to 161 nm (Appendix A). This shift was likely driven by elevated particle concentration skewing the distribution toward larger values. We systematically examined the influence of vortex speed on the size, uniformity, and yield of purified nanogels, excluding speeds that resulted in an incomplete emulsification of the dispersed phase (Appendix A). As vortex speed increased, the mean hydrodynamic diameter and PDI of nanogels declined steadily by 6% and 29.5%, respectively (Figure 4B, Appendix A). Quantitative analysis showed that higher vortex speeds not only suppressed nanogel formation but also elevated the microgel yield, producing a modest shift toward micrometer-scale particles (Appendix A). Increasing vortex speed, i.e., angular velocity, proportionally amplifies the tangential forces within the emulsion. While these enhanced shear forces promote particle breakup, they concurrently increase the collision frequency among emulsion droplets at elevated vortex speeds, thereby facilitating coalescence events that favor the formation of larger microgels. Similarly, prolonged exposure to high vortex speeds shifts the system away from the nanoscopic regime, highlighting the critical role of vortex-induced shear in controlling hydrogel size distributions (Appendix A). Prolonged vortexing led to a significant reduction in PDI, specifically a 54% decrease when vortexing was extended from 30 to 480 s, while simultaneously driving up the average nanogel diameter by approximately 20% (Figure 4C). This behavior mirrors the effects observed under higher vortex speeds and can be attributed to enhanced coalescence: the longer droplets were sheared, the more frequently they collided and merged. Additionally, the overall nanogel concentration rose with extended vortex times, signifying that a greater fraction of material originally in the microgel range was being refined into the nanoscale domain (Appendix A).

The impact of polymer molecular weight on nanogel characteristics was evaluated by comparing formulations prepared with commercially available PEGDA 575 and 700 (Figure 4D). Increasing the polymer weight induced a distinct reduction in the average nanogel diameter and its PDI, by −7% and −19%, respectively. This was further accompanied by a decrease in nanogel concentration. The improvement in mean size and uniformity could be attributed to the PDI of the polymer used, where PEGDA 700 showed a slightly lower PDI compared to PEGDA 575 (Appendix A). An increase in the polymer’s PDI has been associated with the formation of larger and more polydisperse nanoparticles. This relationship could explain why nanogels composed of PEGDA 700 exhibited reduced size and lower PDI, whereas the microgel population showed an increase in both parameters. As illustrated in Figure 2F, employing a longer polymer results in particles with greater size and dispersity. Consequently, the formation of larger microparticles may consume more material, reducing the available polymer for nanogel synthesis and thus accounting for the observed decrease in nanogel concentration when PEGDA 700 was used.

As shown in Figure 4E, increasing the oil-to-polymer ratio produced a gradual reduction in both PDI and mean nanogel diameter, i.e., 62% and 19%, respectively. Nanogel concentration reached a maximum at ratios of 200:50 and 400:50 before declining at 800:50. This trend parallels our observations in the microgel fraction, as intermediate oil levels provided sufficient surfactant coverage to stabilize emulsion droplets, whereas at an 800:50 ratio, the vortex vessel became overfilled. Under these conditions, excessive oil promoted the formation of larger, more heterogeneous microgels, thereby reducing the overall yield of nanogels.

Because overfilling the Eppendorf^®^ vessel compromised emulsion dispersity, we held the oil phase constant and varied the volume of the dispersed polymer phase to adjust the oil-to-polymer ratio (Figure 4F). As the polymer fraction decreased, the mean particle size steadily declined, reflecting the higher surfactant-to-polymer ratio’s enhanced stabilization of smaller nanogels and suppression of coalescence. The lowest polydispersity index (PDI  =  0.04) was achieved at an oil-to-polymer ratio of 400:30. Reducing the polymer phase further (400:10) caused the PDI to surge to 0.20, demonstrating that exceeding the optimal ratio destabilized the emulsion and increased dispersity.

We also varied the crosslink density of the hydrogels to assess whether polymer network density affects particle size and PDI. When increasing the PEGDA concentration from 20 to 30 and 40 wt%, a clear decrease was seen concerning the PDI of the nanogels (−54%), accompanied by a modest 5% reduction in mean diameter (Appendix A; Appendix A). Cryo-TEM further revealed that lowering the PEGDA concentration in the dispersed phase produced a visibly more open polymer network (Figure 5). Additionally, a distinct shift to the nanoscale regime was seen when using 20 wt% PEGDA, where 48% of the distribution is in the nano range, compared to the 14.5% and 19.6% of 30 and 40 wt%, respectively (Appendix A). A likely explanation for this behavior is the viscosity of the dispersed phase. Reducing the PEGDA concentration lowers the viscosity, meaning the phase offers less resistance to the high shear applied during emulsification. More of the input energy is therefore converted into the creation of many small droplets rather than dissipated, allowing for the formation of a large population of nanogels.

In contrast, a higher-viscosity dispersed phase, at 30 wt% and 40 wt% PEGD, resists deformation and breakup more, which could explain the increase in average nanogel size between 30 and 40 wt%, as well as a slight increase in PDI (Appendix A).

To analyze the role of oil viscosity, we substituted n-hexadecane with squalene, a more viscous oil with similar density (0.773 g/mL and 0.858 g/mL, respectively).

Decreasing the viscosity ratio (*η*_D_/*η*_C_) with squalene yielded 36% more microgels and 51% more nanogels (Appendix A). The increase in particle concentration explains the observed upshift in both average size and PDI. In addition, the higher viscosity of the continuous phase enhances emulsion stability and counters droplet coalescence, resulting in more hydrogels being formed from the same starting materials than is achievable with n-hexadecane. The reproducibility of the proposed protocol is further evident in the nanoscale regime, as reflected in the consistent nanogel sizes and dispersities shown in Appendix A.

### 2.3. Ratio of Micro- to Nanogels

We investigated whether particular conditions induce a uniform shift in the entire size distribution toward either the micro- or nanoscale regime (Appendix A). The volume fraction of each regime was calculated using the following formula:(2)$$Volume\;fraction\;(\%)=\frac{4}{3}\times {\;\left(\frac{\mathrm{Mean}\;\mathrm{average}\;\mathrm{diameter}}{2}\right)}^{3}\times particle\;concentration$$

The analysis demonstrated that tuning key formulation and processing parameters directed the hydrogel size distribution into either the micro- or nanoscale regime, in line with previous observations concerning the size and PDI at different conditions. For a shift toward the microscale, the optimal settings were an oil-to-polymer ratio of 50:50, 1% surfactant, 30 wt% PEGDA 700, vortexing at 2400 rpm for 120 s, and using n-hexadecane as the continuous phase. In contrast, achieving a nanoscale distribution required an oil-to-polymer ratio of 400:30, 10% surfactant, 20 wt% PEGDA 575, vortexing at 3000 rpm for 480 s, and employing a higher-viscosity oil such as squalene. The drastic shift towards the nanoscale regime when using squalene was attributed to its increased viscosity (12 cP at 20 °C) [53] compared to n-hexadecane (3.45 cP at 20 °C) [54]. Increasing the viscosity of the continuous phase lowered the viscosity ratio between the dispersed and the continuous phase, which led to smaller droplet sizes [55]. Furthermore, the increased continuous-phase viscosity would stabilize the nanogels more by offering more resistance to droplet coalescence, as the squeezing of the continuous phase between particles was delayed [56,57]. Consequently, the volume fraction within the nanoscale regime increased markedly.

### 2.4. Vortex Versus Microfluidics

Microfluidics is highly regarded for generating hydrogel droplets with a low coefficient of variation, making it an excellent system for producing uniform and precisely controlled droplets. Typical single-channel microfluidic devices operate at rates ranging from 1 to 800 Hz [58], with advanced multi-parallel-channel configurations capable of achieving impressive rates up to 28 kHz [59] or even ≥1 MHz [60,61]. While vortex-based emulsification does not achieve the same coefficient of variation as microfluidics, it offers significantly higher throughput. Specifically, we showed that our system can generate approximately 3.15 × 10^11^ particles within only four min when squalene was used as the continuous phase, corresponding to a droplet generation rate of roughly 1.31 × 10^9^ Hz (Appendix A). This rate is about 151 times higher than the rate of 8.7 MHz achievable by a highly parallelized microfluidic system using 1000 parallel droplet generators [61]. Moreover, although microfluidics systems with increased channel numbers can significantly boost throughput, they simultaneously elevate the risk of channel clogging and operational complexity, including fabrication and maintenance. Thus, while microfluidics excels in precision and consistency, vortex-based emulsification emerges as a viable method for high-throughput hydrogel fabrication with control over size and PDI.

### 2.5. Swelling Ratios of Hydrogels

As previously mentioned, hydrogels are well known for their ability to swell in aqueous environments, a property that is paramount for biomedical applications. To assess this characteristic, we evaluated the swelling behavior of our synthesized hydrogels, as shown in Appendix A. Reports in the literature show that lower-molecular-weight PEG typically yields suboptimal swelling ratios compared to its higher-molecular-weight counterparts. Additionally, an increase in crosslink density is known to further reduce the swelling capacity of the hydrogel network [62]. Consistent with these findings, our highest observed swelling ratio was 5.09 ± 0.01 for 20 wt% PEGDA 575, which is notably below the typical swelling range of 15 to 45 reported for pure PEG-based hydrogels [63].

While these low-MW PEGDA formulations may not be suitable for all applications requiring high swelling capacity, our group did show that pH-responsive hydrogels mainly consisting of PEGDA 575 could be used for the oral delivery of 5-FU [3]. Furthermore, the focus of this study was to systematically investigate the influence of fabrication parameters on droplet size and polydispersity. The insights gained from this work offer a valuable foundation that can be adapted and extended for the use of higher-MW PEGDA or other hydrogel-forming polymers more appropriate for biomedical use.

## 3. Materials and Methods

### 3.1. Materials

All chemicals were used as received unless otherwise specified. Poly(ethylene glycol) diacrylate (PEGDA) (average MW 575/700), photo initiator 2-hydroxy-4′-(2-hydroxyethoxy)-2-methylpropiophenone (Irgacure 2959) (98%), n-Hexadecane oil, and Span^®^ 80 were purchased from Sigma-Aldrich (Amsterdam, The Netherlands). 1H,1H,2H,2H-Tridecafluoro-1-n-octanol (98%) was purchased from TCI (The Netherlands). Eppendorf^®^ tubes were acquired from VWR (Amsterdam, The Netherlands).

### 3.2. Instruments

Ultrapure water was obtained with the help of an Elga Veolia Purelab (Ede, The Netherlands) Chorus 1 purification system (18.2 MΩ) and used in all procedures. Eppendorf^®^ 5430 R was used for centrifugation. An Echo Revolution microscope was used to visualize the microgels. A Fisher brand™ (Landsmeer, The Netherlands) ZX3 vortex mixer was used for vortexing. An Asahi Spectra (Torrance, USA) MAX-303 compact Xenon light source was used for UV curing. The irradiance of this UV light source was determined using a Thorlabs PM400 (Bergkirchen, Germany) optical power meter. Malvern Zetasizer Pro (Almelo, The Netherlands) was used to determine the hydrodynamic radius and polydispersity index of the formed nanogels (refractive index: 1.4; absorption: 0.01).

### 3.3. Methods

#### 3.3.1. Vortex-Generated Hydrogels

All solutions were purged with nitrogen (N_2_) to remove dissolved oxygen (O_2_) for at least 15 min. PEGDA (30% *w*/*w*) was dissolved in Ultrapure water. N-Hexadecane oil was mixed with Span^®^ 80 non-ionic surfactant. Irgacure 2959 was added to the PEGDA solution before purging (1% *w*/*w* final concentration). First, the polymer solution was pipetted in an Eppendorf^®^ tube, followed by the careful addition of n-Hexadecane along the inner side of the Eppendorf^®^ tube to avoid unwanted emulsification prior to vortexing. The Eppendorf^®^ was then vortexed for a desired amount of time. The resulting hydrogel droplets were cured by exposing the emulsion to a focused UV beam (λ = 300–600 nm, 5 min, 50% intensity, 140 mW/cm^2^) (Appendix A). The emulsion was centrifuged down and washed primarily with ethanol and finally with Ultrapure water. The diameter of the particles was measured using images captured by an Echo Revolution microscope and analyzed with FIJI ImageJ software (v1.54f).

#### 3.3.2. Cryo-TEM Imaging

Cryo-TEM imaging was performed using a JEOL Transmission Electron Microscope 2100 at 200 kV with a high-quality Gatan 895 ultrascan 4000 bottom mount camera (4080 × 4080 pixels) incorporated to capture the density of the generated nanogels. TEM grids (Quantifoil, Großlöbichau, Germany) were glow-discharged by a 208-carbon coater (Cressington, Watford, UK). To ensure adequate particles were present for imaging, samples were concentrated by centrifugation whenever necessary. Subsequently, the sample (3.5 μL) was loaded onto the grid, blotted, and finally vitrified through plunging into liquid ethane at 100% humidity with FEI Vitrobot Mark IV (blot time 1.5 s, blot force 2). Finally, all samples were loaded in a 914 high-tilt cryoholder (Gatan, Munich, Germany) and inserted into the microscope for imaging. Data analysis was performed using Fiji ImageJ (v1.54f).

#### 3.3.3. Particle Concentration Calculations

Nanogel concentration was determined using a Nanosight LM10 (Almelo, The Netherlands), equipped with a 643 nm laser (40 mW) and a 20x objective. Five 30 s videos were taken consequently and subsequently analyzed to determine particle concentration. Samples were diluted 20-fold prior to analysis to ensure accurate measurements. Microgel concentration was determined by loading samples into wells formed with SecureSeal™ imaging spacers (Grace Bio-Labs, US) on microscope slides (RS France, Beauvais, France), each of known thickness and diameter. After confirming complete sedimentation of microgels at the bottom of the wells, multiple images were acquired and processed using a custom Fiji ImageJ (v1.54f) labkit model and macro (Appendix A) to extract particle size and count. The resulting counts were then converted to particles per milliliter.

#### 3.3.4. Statistical Analysis

Hydrogel size distribution data were plotted as bar graphs with n = 500, unless otherwise stated. The bars represent the standard deviation of the measured particles; the whiskers indicate outliers; the circle within each box represents the mean; the curve shows the log-normal distribution; and the individual dots correspond to the measured particle diameters. Statistical significance was determined using non-parametric Mann–Whitney and Kruskal–Wallis tests. Statistical analyses were performed on single batches of prepared emulsions, and reproducibility was evaluated across three independent batches.

#### 3.3.5. Viscosity Measurements

The viscosity of the solutions used for hydrogel fabrication was measured using a glass capillary Micro-Ubbelohde viscometer with suspending ball-level. The glass capillary was kept in a water bath at 25 °C. The kinematic viscosity was calculated by the following equation:v = *K* × t
with ‘v’ being the kinematic viscosity in m^2^/s, ‘*K*’ the instrument constant in m^2^/s^2^, and ‘t’ the time in s. From this, the dynamic viscosity can be calculated using the following equation:η = ρ × v
with ‘η’ being the dynamic viscosity in *Pa* × *s*, ‘ρ’ the density in kg/m^3^, and ‘V’ the kinematic viscosity in m^2^/s.

## 4. Conclusions

In conclusion, we have developed a novel and cost-effective method for generating ultra-small and uniform PEGDA hydrogels using a simple tabletop vortex system. This approach streamlined hydrogel synthesis, making it faster, scalable, and adaptable for diverse applications, from advanced drug delivery to tissue engineering. Under optimal conditions, we achieved a coefficient of variation of 0.26 and a PDI of 0.07 in the microscale regime, while achieving a PDI of 0.04 in the nanoscale regime.

Key parameters influencing particle size and PDI across both size regimes include surfactant concentration, vortex duration, vortex speed, and oil-to-polymer ratio, all of which, when increased, contributed to smaller and more uniform droplets. Additionally, a newly developed pulsed vortexing technique further reduced the mean droplet diameter by approximately 15%. However, the use of high-molecular-weight polymers negatively impacted emulsion dispersity in the micro range, resulting in larger droplets and broader size distributions. Similarly, an undersized container, where the vortexed emulsion reached the cap, constrained emulsion dynamics, leading to a decrease in particle uniformity and an increase in mean hydrogel size.

Furthermore, key parameters enabled a pronounced shift in the entire size distribution either towards the micro- or nanoscale regime. We determined that a 50 : 50 oil-to-polymer ratio, 1% surfactant, 30 wt% PEGDA 700, and vortexing at 2400 rpm for 120 s in n-hexadecane yielded an optimally micron-sized hydrogel population, whereas a 400 : 30 oil-to-polymer ratio, 10% surfactant, 20 wt% PEGDA 575, and vortexing at 3000 rpm for 480 s in squalene drove the distribution firmly into the nanoscale. These findings establish clear conditions sets for directing hydrogel architectures across size regimes.

Conventional microfluidic droplet generators deliver outstanding monodispersity, achieving particle generation frequencies of ≥1 MHz in multi-parallel-channel setups. By contrast, our vortex-based emulsification generated ≈ 3.15 × 10^11^ hydrogel droplets in just 4 min—equivalent to 1.31 × 10^9^ Hz—yielding a 150-fold increase in throughput over a 1000-nozzle microfluidic array while avoiding the clogging risk and the operational complexity of massively parallel chips. These results establish vortex emulsification as a practical, high-throughput alternative that still affords tight control over droplet size and polydispersity.

The protocol presented here provides a flexible and adaptable platform for the generation of both micro- and nanogels. The insights gained from this study allow for precise control over key parameters to optimize spherical hydrogel particle synthesis. We believe this method presents a robust, accessible, and rapid alternative to microfluidics for producing sub-5-micron hydrogel particles with low dispersity, requiring only standard laboratory equipment and no specialized training.

## Figures and Tables

**Figure 1 ijms-26-06300-f001:**
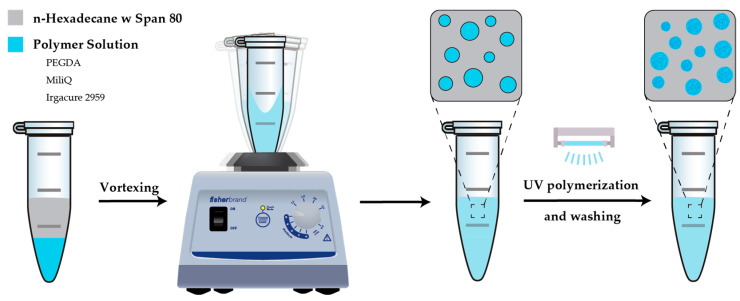
Schematic representation of the simplified vortexing protocol: the dispersed phase (PEGDA in Milli-Q water) is first pipetted slowly into the Eppendorf^®^, followed by the careful addition of the continuous phase (n-hexadecane with Span^®^ 80) on top. The mixture is rigorously vortexed for complete emulsification, followed by UV polymerization to crosslink PEG chains, forming spherical hydrogels that are subsequently washed.

**Figure 2 ijms-26-06300-f002:**
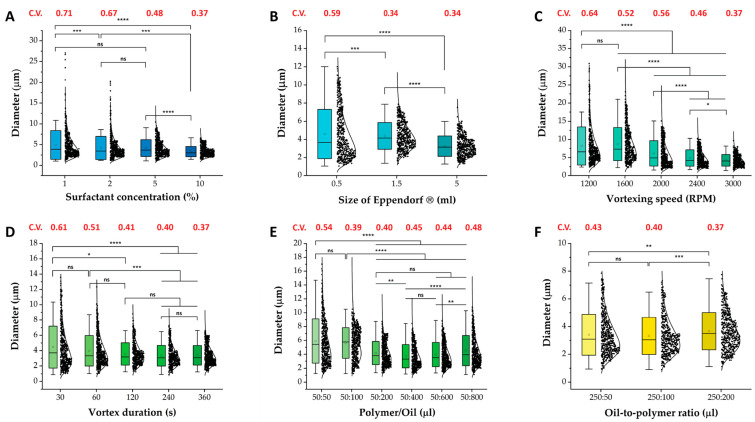
Size distributions of polymerized PEGDA hydrogel particles illustrating the impact of various variables on average particle size and polydispersity index. The investigated variables include surfactant concentration (**A**), Eppendorf^®^ size (**B**), vortexing speed (**C**), vortexing duration (**D**), oil-to-polymer ratio with constant polymer amount (**E**), and oil-to-polymer ratio with constant oil amount (**F**). Increasing surfactant concentration, Eppendorf^®^ size, and vortex speed and duration all result in narrower size distributions. Increasing the oil-to-polymer ratio also positively influences particle uniformity, as long as the Eppendorf^®^ is large enough to accommodate the emulsion. The coefficient of variation (C.V.) shows changes in sample dispersity. The size distributions were analyzed using ImageJ, while statistical analysis was performed using a non-parametric Kruskal–Wallis test, based on a sample sizes of n = 500. Statistical significant differences among variables are shown as *, **, ***, **** and ns (*p* ≤ 0.05, *p* ≤ 0.01, *p* ≤ 0.001, *p* ≤ 0.0001 and *p* ≥ 0.05, respectively). The bar is the standard deviation of the measured particles, the whiskers represent the outliers, the circle within the box signifies the mean, the curve represents the log-normal distribution, and the black dots represent the diameters of the measured particles.

**Figure 3 ijms-26-06300-f003:**
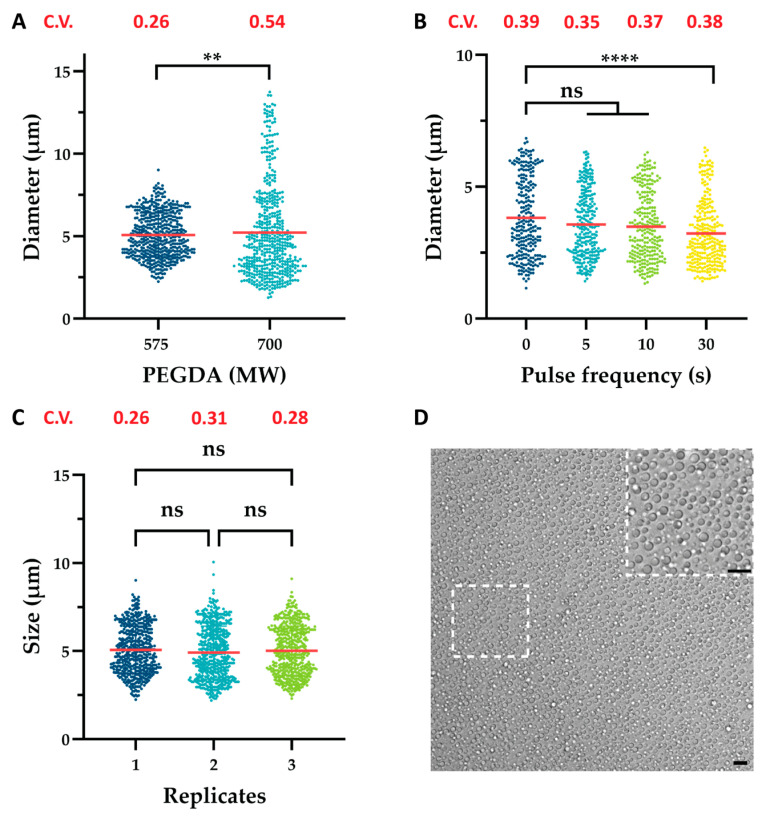
Size distributions of polymerized PEGDA hydrogel particles illustrating the influence of variables on average particle size and polydispersity index. (**A**) Effect of polymer molecular weight, (**B**) effect of pulse frequency, and (**C**) reproducibility in triplicate. (**D**) Representative bright-field image of hydrogel particles synthesized under optimal conditions (5 mL Eppendorf tubes, 10% surfactant concentration, 30 s pulsing, PEGDA 575 MW, 240 s vortexing, polymer-to-oil ratio of 50:300, and vortexing speed of 3000 rpm). The coefficient of variation (C.V.) shows changes in sample dispersity. Size distributions were analyzed using ImageJ software, and statistical comparisons were performed using the non-parametric Mann–Whitney (**A**) and Kruskal–Wallis (**B**,**C**) tests (**A**,**C**): n = 500; (**B**): n = ± 250). Statistical significant differences among variables are shown as **, **** and ns (*p* ≤ 0.01, *p* ≤ 0.0001 and *p* ≥ 0.05, respectively). Red lines signify sample means. Scalebars are 10 µm.

**Figure 4 ijms-26-06300-f004:**
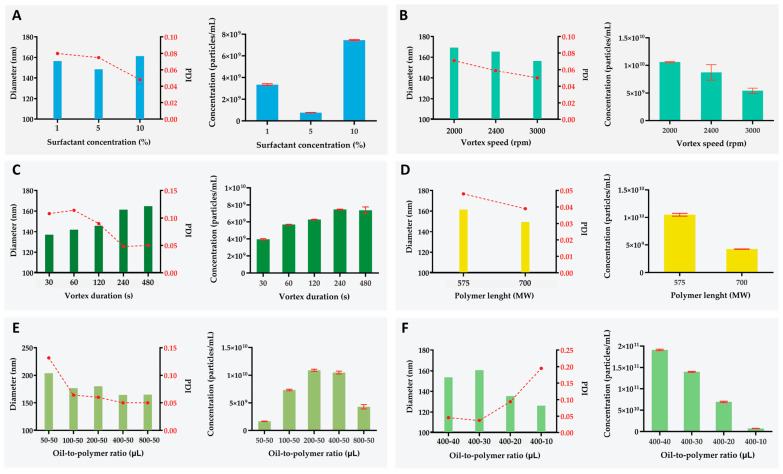
Effect of key experimental variables on the average size and polydispersity index of polymerized PEGDA nanogels: (**A**) effect of surfactant concentration, (**B**) effect of vortex speed, (**C**) polymer molecular weight, (**D**) vortex duration, (**E**) oil-to-polymer ratio, and (**F**) crosslinking density. Left-hand panels show the mean particle diameter (left Y-axis, bars) together with its corresponding polydispersity index (PDI) (right Y-axis, red dots), while right-hand panels display the concentration of nanogels formed under each condition.

**Figure 5 ijms-26-06300-f005:**
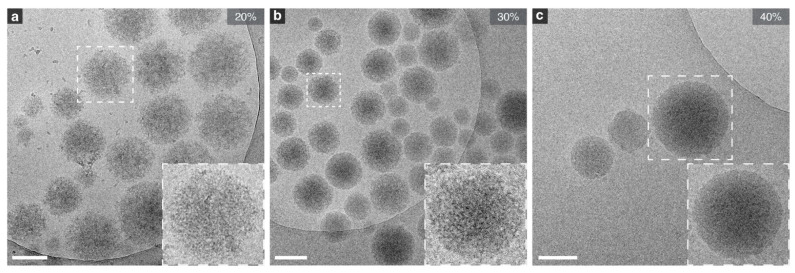
Representative Cryo-TEM images of nanogels with varied crosslinker content: (**a**) 20 wt%, (**b**) 30 wt%, and (**c**) 40 wt%. Network density increases with polymer concentration, from the most open structure at 20 wt% to the densest at 40 wt%. Scale bars represent 200 nm.

## Data Availability

The raw data supporting the conclusions of this article will be made available by the authors on request.

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
