# Peer review of "A Simple Vortex-Based Method for the Generation of High-Throughput Spherical Micro- and Nanohydrogels"

_ijms, 2025, doi:10.3390/ijms26136300_

Round 1
Reviewer 1 Report
Comments and Suggestions for Authors
The article presented, "Simple vortex-based method for the generation of uniform spherical micro and nanohydrogels", is interesting and relevant to the areas of biomaterials. Using vortex mixing and UV photopolymerisation, the study converts PEGDA water‑in‑oil emulsions into uniform micro‑ and nanohydrogels. Adjusting vortex speed, pulsing, Span‑80, oil viscosity and PEGDA mass yields droplets from 4µm to 160nm with PDI≤0.06. Throughput exceedssparticles, matching multi‑nozzle microfluidics while avoiding clogging and complex equipment. Some points can be considered:
First, there are some spelling errors as described below, however I suggest a detailed review
* correct on line 380 "preformed"
*correct on line 392 and 393 "consequently and 392 subsequently"
*correct on line 348 "maitanance"
*exchange "continuous" for "continuous"
Later, answer some general questions:
Do the authors have any rheological measurements? It would be excellent to evaluate the viscous effects of the hydrogel.
The authors "We also identified a critical limitation related to the size of the container used for emulsion generation. Specifically, when the emulsion reached the cap of the vial during vortexing, the emulsification process became inefficient." Is there any limit? And is there any relationship with the filled height since it is an Eppendorf?
On lines 108-208 the author proposes that 30s vortex pulses “reset turbulence” and narrow the size distribution, but no kinetic data are presented. Have you monitored droplet count or torque during pulsing to substantiate this mechanism? If not, please temper the claim or add supporting experiments.
In Figures 2–4, overlay Gaussian curves on size histograms (e.g., Figure3)
. Emulsions typically follow log‑normal statistics (e.g. https://pubs.acs.org/doi/full/10.1021/acs.langmuir.3c02463). Have the authors tested a log‑normal fit or reported skewness to justify the Gaussian assumption?
What are the limits to define what is nano and micro gel?
The UV source is specified as 300–600nm, 5min, “50% intensity”. What irradiance (mWcm⁻²) reaches the sample and how is lamp aging or spatial inhomogeneity accounted for? Consider reporting the total energy dose to facilitate reproducibility.
Higher vortex speed reduces microgel diameter but is also said to suppress nanogel formation and shift product distribution towards microgels. Could the author explain the experimental conditions for the two observations and explain why the trends differ? A table summarizing the parameter matrix (aqueous fraction, RPM, surfactant %) could be checked.
What refractive index and absorption values were used for PEGDA hydrogels in Malvern Zetasizer calculations?
Comments on the Quality of English LanguageThe language is reasonable, but there are some typographical errors in the text that can hinder understanding.
Reviewer 2 Report
Comments and Suggestions for Authors
The manuscript entitled “Simple vortex-based method for the generation of uniform spherical micro and nanohydrogels” (ijms-3693701) represents the preparation of micro- and nano-sized hydrogels by vortexing for potential applications. The manuscript is well-designed, organized, and clearly explained, with results presented in an adequate manner. However, some comments need to be addressed before publication:
- Please include some literature survey of how other researchers attempted to prepare such hydrogel (material-wise and method-wise) for any specific application and present the effect of hydrogel properties on the outcome of the applications.
- Please include the schematic polymerization mechanism.
- Please confirm the synthesis of the hydrogel using analytical techniques such as NMR and FTIR.
- One of the most significant properties of hydrogels is their ability to absorb water. Please study the water absorbance of the synthesized hydrogel and report in “g/g”.
- Please move the TEM images (Fig. S6.) to the main manuscript. They are worth seeing there.
Reviewer 3 Report
Comments and Suggestions for Authors
In the paper entitled "Simple vortex-based method for the generation of uniform spherical micro and nanohydrogels", the authors propose a simple method for the generation of spherical micro and nanohydrogels by using vortex to prepare water in oil emulsions. The authors explore optimized conditions such as surfactant concentration, vessel volume, phase viscosity, vortex speed and duration, and so on... The work is well written and of interest for a large community of scientists. However, some methods and derived results should be better highlighted. Therefore, this reviewer suggests major revisions before it could be published in IJMS journal.
Please find the point-by-point suggestions below:
- The introduction should be significantly extended with the current state of the art concerning simple and low cost micro and nanohydrogel formulations, to highlight the novelty of the proposed approach compared with the already reported ones in the literature.
- The title is misleading: the authors highlight that the proposed method leads to the generation of uniform micro and nanohydrogels, but actually the populations they derive are quite non-uniform, and several centrifugation steps are required to isolate the different populations. Maybe large-scale production (or high throughput) could be more adequate for the proposed work.
- The authors suggest the use of these nano and microhydrogels for drug delivery and tissue engineering. However, it is well known in the literature that PEGDA 700 and 575 have poor swelling capabilities compared to PEGDA with much higher MW (10000 and 20000). Did the auhors perform swelling tests of the polymerized micro and nanogels? How was the PEGDA solution concentration (30 % w/w) chosen?
- The statistical analysis shown in the manuscript seem to have been performed on single batches of hydrogel nano and microspheres. Did the authors perform a batch-to-batch variation analysis? If yes, on how many batches? A statistical analysis paragraph should be added in the methods to describe the number of replicates per condition that the authors have performed.
Round 2
Reviewer 3 Report
Comments and Suggestions for Authors
The authors assessed all the concerns from this reviewer, the manuscript has been improved, and it can now be accepted in the present form.